# Effect of *Plateau pika* on Soil Microbial Assembly Process and Co-Occurrence Patterns in the Alpine Meadow Ecosystem

**DOI:** 10.3390/microorganisms12061075

**Published:** 2024-05-26

**Authors:** Xiangtao Wang, Zhencheng Ye, Chao Zhang, Xuehong Wei

**Affiliations:** 1School of Life Sciences, Guizhou Normal University, Guiyang 550025, China; 2Qiangtang Alpine Grassland Ecosystem Research Station, Tibet Agricultural and Animal Husbandry University, Nyingchi 860000, China; 3State Key Laboratory of Soil Erosion and Dryland Farming on the Loess Plateau, Northwest A&F University, Xianyang 712100, China; yezhencheng1996@nwafu.edu.cn (Z.Y.); zhangchaolynn@163.com (C.Z.); 4Institute of Soil and Water Conservation, Chinese Academy of Sciences and Ministry of Water Resources, Xianyang 712100, China

**Keywords:** mammalian engineer, microbial biogeography, community assembly, alpine ecosystem

## Abstract

Burrowing animals are a critical driver of terrestrial ecosystem functioning, but we know little about their effects on soil microbiomes. Here, we evaluated the effect of burrowing animals on microbial assembly processes and co-occurrence patterns using soil microbiota from a group of habitats disturbed by *Plateau pika*s (*Ochtona curzoniae*). Pika disturbance had different impacts on bacterial and fungal communities. Fungal diversity generally increased with patch area, whereas bacterial diversity decreased. These strikingly different species–area relationships were closely associated with their community assembly mechanisms. The loss of bacterial diversity on larger patches was largely driven by deterministic processes, mainly due to the decline of nutrient supply (e.g., organic C, inorganic N). In contrast, fungal distribution was driven primarily by stochastic processes that dispersal limitation contributed to their higher fungal diversity on lager patches. A bacterial co-occurrence network exhibited a positive relationship of nodes and linkage numbers with patch area, and the fungal network presented a positive modularity–area relationship, suggesting that bacteria tended to form a closer association community under pika disturbance, while fungi tended to construct a higher modularity network. Our results suggest that pikas affects the microbial assembly process and co-occurrence patterns in alpine environments, thereby enhancing the current understanding of microbial biogeography under natural disturbances.

## 1. Introduction

Alpine ecosystems, which typically occur at higher altitudes, above the tree line [1], are among the world’s most important terrestrial ecosystems and play key roles in the maintenance of biodiversity, global carbon, and climate balance [2,3], being of great significance to regional ecological quality and safety. However, such ecosystems are also extremely fragile and highly vulnerable to environmental variation (e.g., climate change and biological disturbance) [4,5] because they characteristically experience low temperature and high UV radiation, have a short growing season, and poor nutrient status [6,7], and depend on mammal species that function as ecosystem engineers [8,9].

In alpine meadows, the burrowing activities of mammals can generate bare patches of soil [10] and can significantly affect soil properties (e.g., nutrient content and porosity), as well as plant diversity and productivity [11]. The patches resulting from such activity are also expected to possess greater ranges of microenvironments (e.g., litter depths and soil nutrient levels) than the surrounding areas [12] and thus greater niche partitioning and richer microbiomes [13,14], which subsequently play critical roles in ecosystem functioning and ecological processes (e.g., cycling of C, N, P, and other minerals). As such, elucidating the effects of mammalian disturbance on soil microbiomes is extremely important for understanding the maintenance of biodiversity and ecosystem functioning, as well as for formulating effective ecosystem management and conservation policies, especially regarding soil biochemistry, and this work is especially important considering the extremely high rates of extinction and range reduction in small-to-medium (50–5000 g) burrowing mammals (e.g., meadow rabbits, grassland squirrels, and short-beaked echidnas) over the past 100 years [15,16].

Interestingly, the bare patches created by burrowing mammals are often island-like due to their discrete nature and distinct soils and vegetation [10], when compared to surrounding areas, and, thus, the microbe diversity of such patches is likely subject to the processes of island biogeography, such as colonization and extinction, and may depend on patch size and habitat heterogeneity [17,18]. Other factors, such as dispersal limitation and environmental selection, may also play major roles in microbe distribution, especially in large areas and/or spatially isolated patches [19,20], whereas homogeneous dispersal and drift may play major roles at smaller spatial scales [19,21]. Indeed, species-area relationships (SARs) have proven useful in explaining microbe species diversity in ‘virtual island’ systems (e.g., water-filled treeholes, cryoconite holes, and oil sump tanks) [17,22,23]. However, little is known about the SARs of island-like microhabitats created by the disturbance of ecosystem engineers.

The burrowing activity of the *Plateau pika* (*Ochotona curzoniae*) and the different-sized patches it generates provide a natural experiment for investigating the SARs of microbiomes in the disturbed microhabitats of an alpine ecosystem. Despite suffering widespread extinction and dramatic population contraction in other regions of the world, the *Plateau pika* has successfully survived in Hengduan Mountains and nearby regions of the Qinghai-Tibetan Plateau [24]. This small burrowing herbivore prefers an open habitat and avoids thick vegetation [25]. Once *Plateau pika*s occupy some suitable alpine meadows, their disturbance will further make these alpine meadows become more open and short-statured [10,26]. The burrowing activity of this small mammal greatly transforms landscapes and increases the proportion of bare ground, thereby promoting micro-topographical heterogeneity [27], which can affect seed germination, litter capture, and rainwater infiltration [25,28], as well as various ecosystem functions (e.g., habitat availability, nutrient cycling, hydrology, and productivity) [29], and may further influence soil microbial communities.

Accordingly, the aims of the present study were to evaluate the effects of a key mammalian ecosystem engineer, the *Plateau pika O. curzoniae*, on the spatial structure (diversity and co-occurrence patterns) of soil fungi and bacteria communities in alpine ecosystems of the Qinghai–Tibetan Plateau and to elucidate any underlying mechanisms. The present study included a standardized field survey of 51 quadrats with six different-sized patches associated with pika digging. More specifically, the study addressed the hypotheses that (1) mammalian disturbance can regulate the diversity and co-occurrence patterns of alpine soil microbiomes, (2) bacterial and fungal communities exhibit positive SARs since the greater heterogeneity of larger patches promotes species diversity, and (3) the SARs of alpine soil microbiomes are mediated by a variety of ecological processes (e.g., deterministic and stochastic processes).

## 2. Materials and Methods

### 2.1. Study Design

The present study was conducted at the Bangjietang alpine meadow experimental station, which is located on the Qinghai–Tibetan Plateau (29°54′48″ to 29°55′33″ N and 92°21′41″ to 92°22′24″ E; 4460–4470 m). This region has a typical continental plateau climate, characterized by low temperature and high humidity, and the mean annual air temperature is −3.1 °C, whereas the annual precipitation is 409 mm, with more than 80% of precipitation occurring during the summer growing season. According to the Chinese Soil Classification System, the soil in this region is classified as alpine meadow soil (similar to Cambisols in FAO/UNESCO classification system), and the dominant plant species are *Kobresia pygmaea*, *Kobresia robusta*, and *Elymus nutans*. The herbivorous *Plateau pika* (*O. curzoniae*) is also common in the region.

In August 2020, a cluster of 6 bare patches generated by pika burrowing activity was selected for study. The size of the 6 patches varies from 4 to 592 m^2^. We used a hierarchical sampling regime to collect soil samples [18,30]. On each patch, we established three to fifteen 0.5 m × 0.5 m quadrat, with the number of quadrats roughly proportional to patch area on the logarithmic (log) scale. For each quadrat on the disturbed patches, five evenly distributed soil cores (5 cm diameter to 20 cm depth) were taken and mixed to form one composite sample, resulting in a total of 51 soil samples. The distance between patches was approximately 50–100 m with no overlap between them. All patches were located in the same alpine meadow and thus were characterized by the same soil type, topography, and microclimate. The composite soil samples for each quadrat were sieved (2 mm mesh) to remove roots and rocks, homogenized, and separated into two parts. One part was stored at 4 °C for soil analysis, and the other was transferred to a sterile plastic bag and frozen (−40 °C) for subsequent DNA analysis. More detailed information about the soil sampling can be found in Appendix A.

### 2.2. Soil Physicochemical Analysis

Soil total nitrogen (TN) was measured using the semimicro-Kjeldahl method and a fully automated Kjedahl analyzer (Kjeltec 8400; FOSS Corporation, Hillerød, Denmark). Total phosphorus (TP) and available phosphorus (AP) in soil were measured using a colorimetric method and a UV-visible spectrophotometer (UV-2550; Shimadzu, Kyoto, Japan). Available potassium (AK) was extracted using NH_4_OAc and then measured using flame photometry. All other soil properties, namely soil moisture, pH, organic carbon (OC), nitrate nitrogen (NO_3_^−^), and ammonium nitrogen (NH_4_^+^), were determined using standard analytical methods, as described by Ye et al. [31].

### 2.3. Soil Microbiome Analysis

Genomic DNA was extracted from soils collected from pika burrowing patches, using a FastDNA SPIN Kit for Soil (MP Biochemicals, Solon, OH, USA) according to the manufacturer’s instructions. Bacterial 16S rRNA and fungal ITS regions were amplified using specific primers (338F/806R and ITS1-1737F/ITS2-2043R, respectively) [32,33] and then sequenced using an Illumina HiSeq2500 platform (Illumina Inc., San Diego, CA, USA). The raw paired-end reads were de-noised, assembled using DADA2 (v1.1.3) [34], clustered into different amplicon sequence variants (ASVs) based on 100% sequence similarity, and then were subject to taxonomic assignment using Ribosomal Database Project classifiers within the SILVA (v132) and UNITE databases for bacteria and fungi, respectively. ASVs that were not classified into bacteria or fungi were removed. The samples were rarefied to an even number of sequences per sample (37,925 and 47,094 for bacteria and fungi, respectively).

### 2.4. Evaluation of Microbe Specialization

Microorganisms can be classified as habitat generalists and specialists, based on their niche breadth [35]. Habitat generalists are considered more resilient, with broad environmental tolerances, whereas habitat specialists are only capable of adapting to specific habitats and possess relatively narrow environmental tolerances. Both bacterial and fungal microbes were designated as habitat generalists or specialists based on Levins’ niche breadth index, which was calculated using EcolUtils (v0.1; https://github.com/GuillemSalazar/EcolUtils, accessed on 15 January 2024) with 1000 permutations. ASVs with a high Levins’ niche breadth occur at evenly distributed abundances along with a wide range of habitats, and can therefore be classified as habitat generalists. Meanwhile, ASVs with a low Levins’ niche breadth have uneven distributed abundances among habitats, and can therefore be considered as habitat specialists [35].

### 2.5. Evaluation of Microbiome Assembly Processes

The effects of stochastic processes on community assembly were assessed using the neutral model described by Sloan et al. [36], which predicts the relationship between taxon frequency and abundance within a set of local communities. In this model, migration rate (m) represents the dispersal capacity, with higher values indicating less dispersal limitation, and, in the present study, this rate was calculated using the mean relative abundance of observed ASVs. The R^2^ value of the neutral community model (NCM) represents the goodness of the fit of the model, with R^2^ values approaching 1 indicating that a community’s assembly is driven entirely by stochastic processes.

Null model analysis described by Stegen et al. [21] was also performed to investigate the proportional contributions of variable selection, homogeneous selection, dispersal limitation, homogeneous dispersal, and ecological drift to the assembly process. The null model expectation was generated using 999 randomizations. The β-nearest taxon index (βNTI) and Bray–Curtis-based Raup–Crick (RC_bray_) metrics were calculated using null model-based phylogenetic and taxonomic β-diversity metrics to measure differences in both phylogenetic diversity and taxonomic diversity. For a given community, |βNTI| > 2 indicated that the community was driven by deterministic processes, whereas |βNTI| < 2 indicated that the community was driven by stochastic processes. Furthermore, the relative contribution of variable selection was quantified as βNTI > 2, whereas the relative contribution of homogeneous selection was identified as βNTI < −2. The relative contributions of dispersal limitation were quantified as the percentage of pairwise comparisons with |βNTI| < 2 and RC_bray_ > 0.95, whereas homogenizing dispersal was quantified as the percentage of pairwise comparisons with |βNTI| < 2 and RC_bray_ < −0.95. In addition, the fractions of all pairwise comparisons with |βNTI| < 2 and |RC_bray_| < 0.95 were used to assess the impact of “undominated” assembly.

### 2.6. Microbial Co-Occurrence Network Construction

Co-occurrence networks based on correlations between taxon occurrence were constructed to estimate species coexistence patterns [37]. For the bacterial network, only ASVs present in at least half of the samples were included [38], and co-occurrence was evaluated according to Spearman’s correlation coefficient (ρ > 0.8, *p* < 0.01). For the fungal network, only ASVs present in at least 40% of the samples were used, and co-occurrence was evaluated according to Spearman’s coefficient (ρ > 0.5, *p* < 0.05). The network nodes represented ASVs, and the edges indicated strong and significant correlations. Subnetworks were further extracted for each sample to calculate subnetwork parameters, including the total number of nodes (network size) and edges (network connectivity), which essentially reflect network complexity [38], using R v4.1.2 (http://www.r-project.org/, accessed on 15 January 2024).

### 2.7. Statistical Analysis

One-way ANOVA, followed by a post hoc least significant difference comparison, was used to evaluate the significance of differences in the soil variables and microbial α-diversity of patches, with *p* < 0.05 indicating statistical significance. Habitat heterogeneity was measured as the average pairwise Euclidean distances among samples, based on the nine soil properties. α-diversity was defined as the Shannon–Wiener index values, and β-diversity of the microbiome was estimated based on pairwise comparison (i.e., Bray–Curtis dissimilarity) among samples within the patch. We also calculated other diversity indices for α-diversity (i.e., ASV richness and Chao1 index values), and they all yielded similar trends with the Shannon–Wiener index as patch area increased (Appendix A). Linear regression was used to examine whether the SAR, a tendency to enhance diversity from small islands to large islands seen in many higher animals and plants [39], applies to microbial distribution patterns (α- and β-diversity) on patches engineered by small mammals. Random forest analysis was further used to estimate the effects of patch area and soil properties on microbial α- and β-diversity. Furthermore, Levins’ niche breadth index was calculated for soil bacteria and fungi to estimate the proportional influence of stochastic and deterministic processes on soil microbial communities. All statistical analyses were performed in R (version 4.2.1; R Foundation for Statistical Computing; Vienna, Austria.).

## 3. Results

### 3.1. Soil Physiochemistry

Most of the physicochemical soil properties were significantly affected by patch size (Table 1), with TN, OC, AK, AP, inorganic N (NH_4_^+^, NO_3_^−^), moisture, and habitat heterogeneity decreasing significantly with increasing patch area (Appendix A), whereas the variations of soil pH and TP content had no significant correlations with patch area.

### 3.2. Soil Microbiome Composition and Diversity

After rarefying all samples to the same number of sequences, 18,922 bacterial ASVs and 2810 fungal ASVs were retained. At the phylum level, the bacterial communities were dominated by *Proteobacteria* (20.9–28.8%), *Actinobacteriota* (13.1–23.0%), *Acidobacteriota* (15.5–19.2%), *Verrucomicrobiota* (7.1–16.9%), and *Chloroflexi* (9.7–14.1%), and the fungal communities were dominated by *Ascomycota* (0.4–58.3%), *Mortierellomycota* (0.4–36.4%), and *Basidiomycota* (0.1–18.7%; Figure 1). Different diversity patterns were observed for bacterial and fungal microbes, as well as for generalist and specialist microbes. In contrast to typical SARs, bacterial α-diversity was negatively correlated with patch area, and no clear relationship was observed for bacterial β-diversity (Figure 2a,b), whereas both fungal α- and β-diversity were positively correlated with patch area, although only the correlation with fungal β-diversity was significant (Figure 2c,d). Meanwhile, generalist microbe diversity (β-diversity for bacteria and α-diversity for fungi) was positively and significantly correlated with patch area, whereas specialist microbe diversity (β-diversity for both bacteria and fungi), in contrast to typical SARs, was negatively and significantly correlated with patch area (Appendix A).

### 3.3. Drivers of Soil Microbiome Diversity

Random forest modeling indicated that soil properties, such as SOC, NH_4_^+^, and TN levels, were major drivers of bacterial α-diversity, accounting for ~28.8% of the observed variation, but not bacterial β-diversity, for which patch area and soil variables accounted for only ~12.9% of the observed variation (Figure 2e,f). Linear regression also indicated that soil SOC, NH_4_^+^, and TN levels were significantly and positively correlated with bacterial α-diversity (*p* < 0.05; Appendix A). Meanwhile, fungal β-diversity was mainly driven by patch area, which had the highest importance for observed variation, and, to a lesser extent, by soil variables (Figure 2f). The soil properties only accounted for ~1.9% of variation in fungal α-diversity.

The results of random forest modeling showed that the diversity of both bacterial and fungal generalists was mainly driven by patch area and, to a much lesser extent, by soil variables (Appendix A), whereas the patterns observed for specialists were more complex. More specifically, the β-diversity of bacterial specialists was most dramatically influenced by soil habitat heterogeneity, followed by patch area (Appendix A), whereas the β-diversity of fungal specialists was most dramatically influenced by soil AP content, followed by patch area (Appendix A). Linear regression also indicated that habitat heterogeneity and soil AP were significantly and positively correlated with the β-diversity of bacterial and fungal specialists, respectively (Appendix A). These results suggest that the assembly of bacterial and fungal communities was likely driven by different mechanisms (e.g., deterministic and stochastic processes).

### 3.4. Microbiome Assembly Processes

To elucidate the mechanism underlying the distribution patterns of microbial diversity, the extent to which deterministic processes (environmental selection) and stochastic processes (dispersal limitation and drift) drive soil microbial diversity were evaluated. The neutral community and null models indicated that bacterial diversity was mainly driven by deterministic processes, although the relative contribution of deterministic processes decreased gradually with increasing patch area (Figure 3a–h and Appendix A). Meanwhile, fungal diversity was mainly driven by stochastic processes (ecological drift and dispersal limitation; Figure 4a–c), although the relative contribution of the stochastic processes, too, decreased gradually with increasing patch area (Appendix A). In addition, the diversities of both generalist and specialist microbes were mainly driven by stochastic processes, and the effect of deterministic processes was observed to have more impacts on specialist taxa than generalist taxa (Appendix A).

### 3.5. Microbial Co-Occurrence

Bacterial and fungal co-occurrence networks were inferred using correlation relationships. The size (total number of nodes) and connectivity (total number of links) of the bacterial network increased gradually with the enhancement of patch area and were 663.5 and 5068.8% greater, respectively, in the largest patch than in the smallest patch (Figure 5). Variations in the bacterial generalist and bacterial specialist network features (i.e., network size and connectivity) were consistent with those of the bacterial network (Appendix A). However, neither the size nor connectivity of the fungi networks (entire, generalist, or specialist) exhibited any clear relationships with patch area (Figure 5 and Appendix A). To determine the mechanisms by which mammalian disturbance affected microbial network complexity, subnetworks for each patch were generated, and changes in network topological parameters were regressed against patch area. Linear regression indicated that the nodes and edges of the bacterial subnetworks (entire, generalist, and specialist) were positively and significantly correlated with patch area (*p* < 0.001; Appendix A), whereas the nodes of the fungal subnetworks (generalist, and specialist) and the edges of the fungal subnetworks (entire, and specialist) had no significant correlations with patch area. However, the modularity of the fungal subnetworks (entire, generalist, and specialist) was positively and significantly correlated with patch area (*p* < 0.01), while the modularity of the bacterial specialist subnetwork had no significant correlations with patch area.

## 4. Discussion

In the present study, *Plateau pika*-engineered patches of the Qinghai–Tibetan Plateau were used as a model to assess the response of alpine soil microbiomes to mammalian disturbance. Soil microbial diversity, co-occurrence networks, and assembly processes were estimated, and the suitability of typical SARs for describing microbial diversity distribution patterns was evaluated. In contrast to our hypothesis, bacterial diversity and patch area were significantly negatively related, and it is possible that the lower bacterial α-diversity of large patches resulted from poor habitat quality (e.g., low soil SOC, NH_4_^+^, and TN). In other words, larger patches, which suffered greater mammalian disturbance, provided poorer habitat and, thus, harbored lower bacterial diversity. These findings suggest that environmental selection is more likely than patch area to affect soil bacterial diversity [39,40,41]. The results of the neutral community and null models, which indicated that alpine soil bacterial communities are mainly driven by deterministic processes, also confirmed this aspect. Consistent with these findings, Zhang et al. [42] reported that OC and N contents were the most important factors affecting the changes in bacterial communities during long-term grazing exclusion in semi-arid grasslands. However, Bell et al. [17] reported that the size of water-filled treeholes strongly and positively influenced bacterial diversity, possibly because treeholes are formed naturally, whereas pika patches are formed by external disturbances, which may have a dramatic impact on species distribution. Taken together, the results of the present study indicate that habitat quality, especially in regard to soil OC and N contents, is an essential driver of soil bacterial diversity in mammal-engineered patches. Notably, the relative contributions of deterministic processes to bacterial diversity were negatively correlated with patch area, possibly due to reduced habitat heterogeneity. The fragmented vegetation resulting from mammalian disturbance on large patches may buffer the habitat heterogeneity, and stochastic processes may overwhelm deterministic processes in systems with lower environmental variation [37,43]. Indeed, Langenheder et al. [44], in their microcosm experiment, also reported that the local habitat, which did not differ from each other, may weaken the species-sorting effect. However, it is also possible that the reduced contributions of deterministic processes in large patches resulted from greater drift in communities with low bacterial diversity (Figure 3h) [45], which can promote stochasticity under strong environmental selection.

The results of the present study showed that alpine soil fungal communities are driven by stochastic processes and that the positive SAR observed for fungi was mainly driven by patch area. These findings highlight the importance of dispersal limitation in modeling fungal communities within patches. Furthermore, the m value of fungal microbes was at least two orders of magnitude lower than that of bacteria, which also suggested that fungi are more significantly affected by dispersal limitation (Figure 3a–f and Figure 4a) [46]. Consistent with these findings, previous studies have also observed a positive SAR effect in eukaryotes [47] and have reported that fungal communities are more affected by dispersal limitation than bacteria [48,49]. The relatively greater effects of dispersal limitation in fungal communities can be attributed to two mechanisms. First, taxon-specific constraints of fungi may affect long-distance dispersal. For instance, the fungi generally have greater size than bacteria in nature, which may make fungi more vulnerable to suffer from dispersal limitation [50]. Second, even though aeolian processes could transport fungal spores among patches, the patches’ fragmented vegetation might lower wind velocity and thus reduce spore dispersal, especially in larger patches, which contain more fragmented vegetation [51,52]. Also, lower soil moisture could be one reason for the greater effects of dispersal limitation on fungal communities due to the preference of most fungi to humidity. Indeed, the null models indicated that the contribution of dispersal limitation to the fungal communities increased with patch area (Figure 4c), and there was no dispersal limitation in the smallest patch (~4 m^2^), in which ecological drift was the main driver of fungal diversity. This may be because most fungi are capable of dispersing spores over short distances (i.e., centimeters to meters) [53] and because ecological drift is more important in small communities under weak environmental selection [54]. In addition, compared to the study of Adams et al. [55], which reported that the dispersal limitation of fungi occurred within a scale of less than a square kilometer, the present study suggested that soil fungi can be dispersed at relatively finer scales (~56 m^2^). Contrary to the present study, previous studies have reported that environmental selection is the only driver of fungal diversity and that dispersal limitation contributed more to bacterial diversity than environmental selection [56], possibly due to differences in study scale and the strength of environmental gradients in study areas.

The complex interactions that occur within microbiomes can be partially revealed by the generation of co-occurrence networks [57]. In the present study, the network complexity (i.e., size and connectivity) of soil bacteria increased with patch area, whereas networks of soil fungi exhibited high modularity to resist mammalian disturbance. The increased complexity of the bacterial network could be due to the homogeneous soil of large patches, which could weaken niche differentiation and thus promote stronger interactions [58]. In addition, the fungal network was less complex than the bacterial network, possibly because the fungal community was less influenced by environmental selection [59]. In general, networks with greater connectivity are considered to be linked to the quick responses of microbes to environmental perturbations [60], whereas networks with higher modularity are considered to be better organized as they have more functionally correlative members [61] which can form functional units to implement specific functions such as nutrient acquisition. High microbiome modularity may prevent ambient disturbance from spreading to other modules and thus results in more resistant and robust networks [62]. Thus, a likely explanation for the different responses of bacterial and fungal networks to mammalian disturbance is that bacteria with high variability and growth rates respond faster to environmental disturbance, whereas fungi with strong adaptability in poor soil exhibit greater resistance and resilience to disturbance [63,64]. In addition, bacteria, but not fungi, can fix C and atmospheric N [65]. Thus, fungi might form more functional modules to decompose complex molecules, thereby increasing nutrient acquisition abilities in poor soil [66]. Nevertheless, it is worth noting that network analyses are unable to determine whether niche sharing is accidental or due to functional interactions between different taxa (e.g., in the same module) [67].

The findings of the present study also indicate that the diversities of habitat generalists and specialists are mainly driven by stochastic processes and that deterministic processes contribute more to the diversity of specialists than to that of generalists. Thus, the positive SAR of generalists may be related to greater dispersal limitation, whereas the negative SAR of specialists may be determined collectively by both deterministic and stochastic processes. The lower soil AP content and habitat heterogeneity of larger patches may have reduced the diversity of habitat specialists (selection effect), and the diversity may have also been reduced by the more pronounced drift process (Appendix A). Previous studies have also reported that habitat generalists are strongly driven by neutral processes and that species sorting is more important for habitat specialists than generalists [19,68,69]. This discrepancy between generalist and specialist community assembly may have arisen because habitat generalists possess wider niche breadths and are less influenced by environmental factors [70], whereas habitat specialists are only able to adapt to very specialized environmental conditions [71]. Notably, even though the generalist communities in the present study were driven by stochastic processes, the dispersal limitation played key roles in the bacterial generalist community, whereas the ecological drift mainly influenced the assembly of fungal generalist community (Appendix A). This discrepancy may have arisen because fungal generalists form more functional units (higher modularity) than bacterial generalists, potentially generating functional redundancy and increasing susceptibility to drift [54].

The present study investigated the effects of mammalian disturbance on alpine soil microbiomes. We hypothesized the possible changes of bacteria and fungi under mammalian disturbance conditions. First, the diversity of the bacterial community is mainly driven by deterministic processes, and the diversity decreases with increasing patch area, which is, in turn, inversely related to soil habitat quality. Second, the diversity of the fungal community is mainly driven by stochastic processes, and the diversity increases with increasing patch area as a result of increasing dispersal limitation. In addition, the bacterial and fungal communities resist mammalian disturbance through enhanced network complexity and modularity, respectively.

The results of the present study provide novel evidence for the regulation of soil microbiome structure by mammalian disturbance. More specifically, the negative SAR of the bacterial community suggested that mammalian burrowing, to an extent, threatens soil bacterial diversity and, thereby, the diversity of bacteria in alpine soils could be protected by reducing or preventing mammal-induced habitat degradation, whereas the positive SAR of the fungal community suggested that mammalian burrowing promotes soil fungal diversity, and, thereby, the diversity of fungi in alpine soils could be protected by maintaining large pika-engineered patches. In addition, the distinct patterns of the bacterial and fungal communities were driven by different mechanisms. For example, the inferior habitat quality of larger patches was responsible for the negative relationship between patch area and bacterial α-diversity, whereas the greater dispersal limitation in larger patches was responsible for the greater β-diversity of fungi. Collectively, these findings suggested a novel SAR for alpine soil microbiomes that defy the generally accepted view that bacteria and fungi should exhibit similar biogeographic patterns. The results of the present study also have important implications for the conservation of soil bacterial and fungal diversity; however, the generalizability of these findings should be further evaluated in other ecosystems.

## Figures and Tables

**Figure 1 microorganisms-12-01075-f001:**
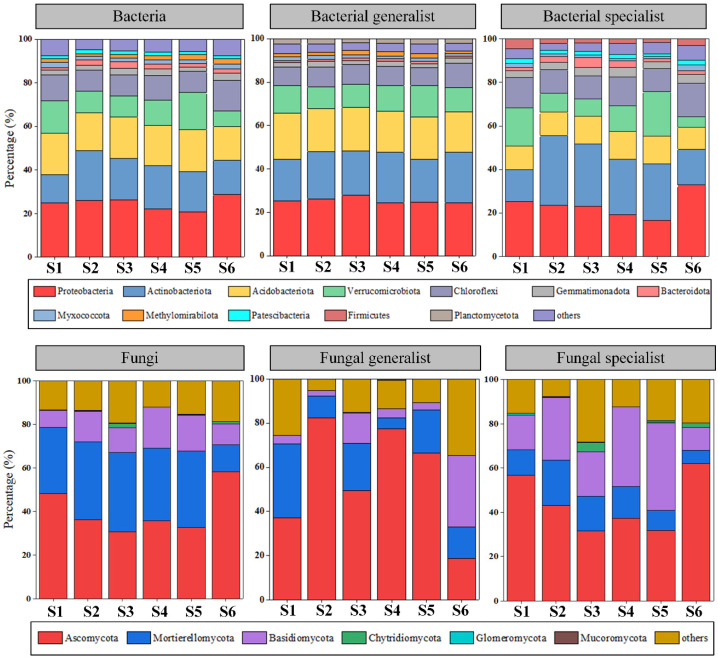
Relative abundance of soil bacteria, fungi, and their generalists and specialists at phylum level on different patches. S1–S6 represents patches ranging in size from small to large.

**Figure 2 microorganisms-12-01075-f002:**
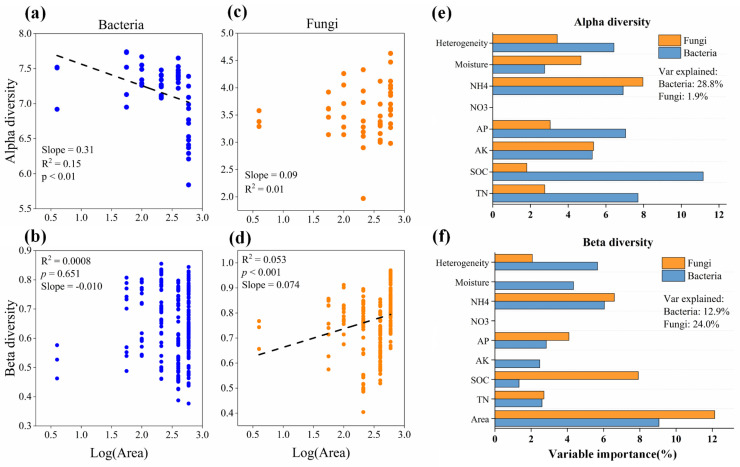
The effects of patch area on the alpha and beta diversity of soil bacteria and fungi, and their important predictors. Panels (**a**,**b**) are for bacterial alpha and beta diversity, respectively; panels (**c**,**d**) are for fungal alpha and beta diversity, respectively. Alpha diversity was measured as the Shannon–Weiner index. Beta diversity was measured as average pairwise Bray–Curtis dissimilarities among samples. Panels (**e**,**f**) are for the fraction of the variation in alpha and beta diversity of soil bacteria and fungi explained by environmental and spatial predictors. TN: total nitrogen; AP: available phosphorus; SOC: soil organic carbon; AK: available potassium; NO_3_^−^: nitrate nitrogen; NH_4_^+^: ammonium nitrogen; Heterogeneity: the habitat heterogeneity, measured as measured as Euclidean distances of these soil properties among samples.

**Figure 3 microorganisms-12-01075-f003:**
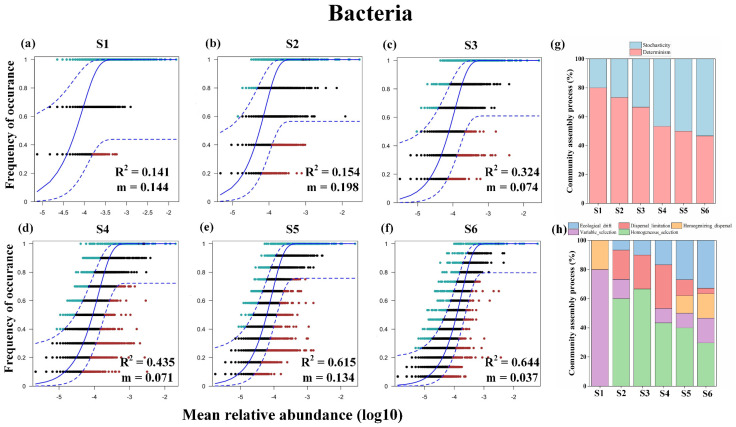
Ecological processes shaping the soil bacterial community assembly. Panels (**a**–**f**) are for the fit of the neutral community model (NCM) of community assembly. The estimated migration rate (m) is a measure of dispersal limitation, and higher m values indicate that microbial communities are less dispersal limited. The R^2^ value of NCM indicates the goodness of fit to the model. Panel (**g**) is for the relative contributions of deterministic and stochastic processes governing bacterial community assembly; panel (**h**) is for the relative contributions of different ecological processes driving bacterial community assembly. S1–S6 represents patches ranging in size from small to large.

**Figure 4 microorganisms-12-01075-f004:**
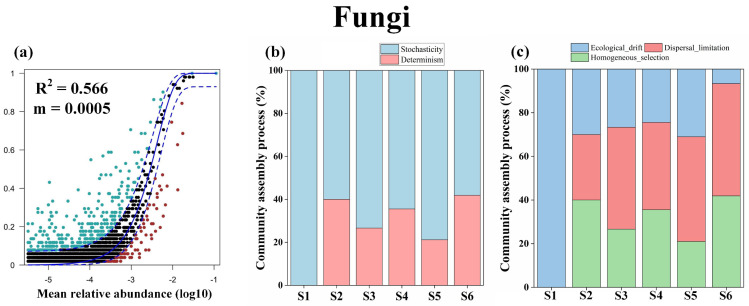
Ecological processes shaping the soil fungal community assembly. Panel (**a**) is for the fit of the neutral community model (NCM) of community assembly. The estimated migration rate (m) is a measure of dispersal limitation, and higher m values indicate that microbial communities are less dispersal limited. The R^2^ value of NCM indicates the goodness of fit to the model. Panel (**b**) is for the relative contributions of deterministic and stochastic processes governing fungal community assembly; panel (**c**) is for the relative contributions of different ecological processes driving fungal community assembly. S1–S6 represents patches ranging in size from small to large.

**Figure 5 microorganisms-12-01075-f005:**
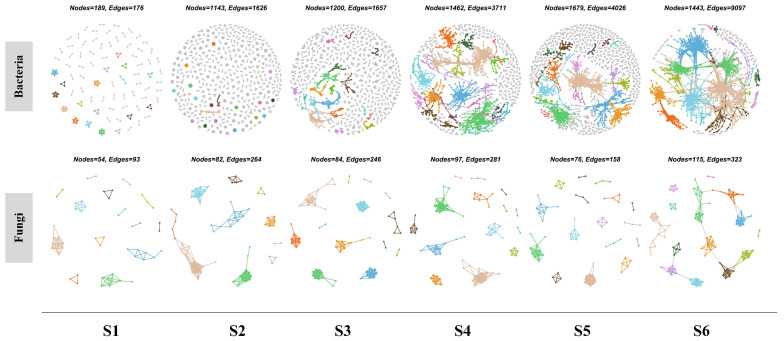
Co-occurrence patterns in soil bacterial and fungal communities on different patches. The top 18 modules in size are shown in different colors, and smaller modules are shown in grey. S1–S6 represents patches ranging in size from small to large.

**Table 1 microorganisms-12-01075-t001:** The soil properties in different patches. Each mean in the Table has an associated SE. The lowercase letters indicate significant differences between the different patches at the 0.05 level of the least significant difference (LSD) test. TN: total nitrogen; TP: total phosphorus; AP: available phosphorus; SOC: soil organic carbon; AK: available potassium; NO_3_^−^: nitrate nitrogen; NH_4_^+^: ammonium nitrogen; Heterogeneity: the habitat heterogeneity, measured as Euclidean distances of these soil properties among samples. S1–S6 represents patches ranging in size from small to large.

	TN g/kg	TP g/kg	SOC g/kg	AK mg/kg	AP mg/kg	NO_3_^−^ mg/kg	NH_4_^+^ mg/kg	pH	Moisture%	Heterogeneity
S1	3.32 ± 0.09 a	0.41 ± 0.01 ab	88.7 ± 3.1 a	70.5 ± 6.9 bc	1.25 ± 0.18 ab	9.9 ± 0.8 b	72.2 ± 6.4 ab	5.79 ± 0.06 a	37.5 ± 2.2 a	25.0 ± 2.4
S2	3.06 ± 0.12 a	0.42 ± 0.01 ab	81.7 ± 2.8 ab	88.6 ± 4.7 a	1.09 ± 0.09 ab	19.7 ± 8.9 a	66.3 ± 6.3 ab	5.76 ± 0.04 ab	33.3 ± 1.3 ab	36.6 ± 4.3
S3	2.94 ± 0.15 a	0.40 ± 0.01 b	76.4 ± 4.1 b	74.4 ± 5.1 b	0.97 ± 0.14 bc	7.8 ± 1.7 b	64.4 ± 4.4 ab	5.78 ± 0.04 a	32.4 ± 1.0 b	26.8 ± 2.3
S4	2.40 ± 0.17 b	0.37 ± 0.00 c	65.2 ± 4.5 c	56.8 ± 3.1 cd	0.31 ± 0.04 d	4.6 ± 0.5 b	59.4 ± 4.1 b	5.63 ± 0.05 b	27.5 ± 1.1 d	28.3 ± 2.0
S5	2.99 ± 0.07 a	0.35 ± 0.01 d	77.1 ± 1.2 b	76.5 ± 4.0 ab	0.40 ± 0.06 d	6.6 ± 0.6 b	75.0 ± 3.5 a	5.51 ± 0.04 c	30.5 ± 0.7 bc	24.0 ± 1.6
S6	2.54 ± 0.05 b	0.43 ± 0.00 a	65.9 ± 1.1 c	54.5 ± 1.6 d	0.77 ± 0.06 c	6.5 ± 0.7 b	43.0 ± 2.3 c	5.81 ± 0.03 a	29.2 ± 0.8 cd	16.5 ± 0.7

## Data Availability

The original data generated in this study are included in this article. Further enquiries can be directed to the corresponding author.

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
