# Peer review of "Effect of Plateau pika on Soil Microbial Assembly Process and Co-Occurrence Patterns in the Alpine Meadow Ecosystem"

_microorganisms, 2024, doi:10.3390/microorganisms12061075_

Round 1

Reviewer 1 Report

Comments and Suggestions for Authors

 The authors did a metagenomic work analyzing the microbial community of areas disturbed by plateau pika, focusing on bacterial and fungal community which were analyzed on the basis of 16S and ITS sequences. In general, the work is well written and can be read easily being and interesting topic. Especially the introduction is well written. The MM section should be clearer with respect to what the hypothesis are how will be validated or rejected, and more detailed with details which should be written and not only referred to the cited works. In the whole manuscript there are some affirmations that are categoric in excess or are not taking into account other scenarios, specially in the discussion. I am in disagree in several sentences/paragraphs of the discussion which should be rewrite some times moving from words like “demonstrated” to “suggests”, sometimes discussing the results in the way of different possible scenarios instead of assuming that the proposed by the authors is the only possible truth. Authors should consider other different explanations for the stochastic distribution and not only dispersal limitations.

My major concerns are the use of different DNA sequences for bacterial and fungal communities, since the 16S is a gene which is expressed and the mutations are limited being a more conserved region than the ITS which is a highly polymorphic marker which doesn´t express. To my understanding it is a big bias. The authors didn´t justified the selection of different markers, and didn´t explained the consequences (possible bias) which can occasionate. To my opinion is a wrong design of the work, even if previous works published in prestigious Q1 journals used the same methodology. My second major concern is the absence of negative controls, the authors used 51 samples from quadrats within patches, however no results neighbor areas are presented.

Find aditional comments attached in a word file.

Reviewer 2 Report

Comments and Suggestions for Authors

This is an outstanding study that offers a comprehensive comparison of bacterial and fungal changes in association with burrowing and resulting bare soil patches from the activities of the plateau pika. The methods used are sound and thorough, and the presentation of data in the manuscript strongly support the conclusions of the authors. The only suggestion pertains to the discussion and the potential role, if any, of microclimatic differences associated with patch size. The authors mention the potential role of wind dispersal with fungi but I am curious if differences in relative humidities that might also explain some of the difference in response between bacteria and fungi. 

Comments on the Quality of English Language

This is, in my opinion, a well written paper. I did note some problems with the language in the abstract, and I went ahead and exported the pdf to a Word document so I could offer suggestions through track changes (that document is upload). Besides the abstract, I noted some irregularities in style among the references cited, and I suggested corrections for these in the attached Word document.
